

# Genome-wide *in silico* identification of membrane-bound transcription factors in plant species

Shixiang Yao, Lili Deng and Kaifang Zeng

College of Food Science, Southwest University, Chongqing, China

## ABSTRACT

Membrane-bound transcription factors (MTFs) are located in cellular membranes due to their transmembrane domains. In plants, proteolytic processing is considered to be the main mechanism for MTF activation, which ensures the liberation of MTFs from membranes and further their translocation into the nucleus to regulate gene expression; this process skips both the transcriptional and translational stages, and thus it guarantees the prompt responses of plants to various stimuli. Currently, information concerning plant MTFs is limited to model organisms, including *Arabidopsis thaliana* and *Oryza sativa*, and little is known in other plant species at the genome level. In the present study, seven membrane topology predictors widely used by the research community were employed to establish a reliable workflow for MTF identification. Genome-wide *in silico* analysis of MTFs was then performed in 14 plant species spanning the chlorophytes, bryophytes, gymnosperms, monocots and eudicots. A total of 1,089 MTFs have been identified from a total of 25,850 transcription factors in these 14 plant species. These MTFs belong to 52 gene family, and the top six most abundant families are the NAC (128), SBP (77), C2H2 (70), bZIP (67), MYB-related (65) and bHLH (63) families. The MTFs have transmembrane spans ranging from one to thirteen, and 71.5% and 21.1% of the MTFs have one and two transmembrane motifs, respectively. Most of the MTFs in this study have transmembrane motifs located in either N- or C-terminal regions, indicating that proteolytic cleavage could be a conserved mechanism for MTF activation. Additionally, approximately half of the MTFs in the genome of either *Arabidopsis thaliana* or *Gossypium raimondii* could be potentially regulated by alternative splicing, indicating that alternative splicing is another conserved activation mechanism for MTFs. The present study performed systematic analyses of MTFs in plant lineages at the genome level, and provides invaluable information for the research community.

Corresponding author
Kaifang Zeng,
zengkaifang@hotmail.com,
zkaifang@swu.edu.cn

## INTRODUCTION

Transcription factors play a primary regulatory role in gene transcription and thus ensure normal growth of plants and promote their adaptation to environmental stress (*Li et al., 2011*). Transcription factors are tightly regulated at multiple levels, including the transcriptional, translational, and post-translational levels (*Seo, Kim & Park , 2008*).

During the last decade, a novel mechanism of post-translational regulation of transcription factors–proteolytic processing dependent activation–has been extensively studied and well established (*Seo, 2014*). A small proportion of transcription factors containing transmembrane (TM) motifs are translated in the cytoplasm and then rapidly anchored to cellular membranes including plasma, mitochondrial, endoplasmic reticulum (ER) membranes, rather than being translocated to the nucleus (*Kim et al., 2010*). Membrane-bound transcription factors (MTFs) are stored in dormant forms and become active after being released from membrane. Most MTFs with known functions have a single TM domain located adjacent to their transcription factor domains located within either the N-terminal or C-terminal region and they are anchored to the membrane by TM motifs. Thus, in response to environmental stimuli and physiological signals, the MTFs can be cleaved near the TM region by a specific protease, and the transcription factor can be released from the membrane and then translocated into the nucleus to exert its functions (*Che et al., 2010*). The proteolytic processing-dependent activation of MTFs skips both transcriptional and translational steps, and it thus ensures the prompt response of plants to exogenous and endogenous signals.

Since the first plant MTF, *AtbZIP60*, was elucidated at the molecular level, a major focus of research on the MTFs of plants has targeted NAC and bZIP transcription factors (*Iwata & Koizumi, 2005*; *Seo, 2014*). For instance, the function of several MTFs has been well elucidated in recent years, including eight NAC and three bZIP members (*Seo, 2014*). Previous studies of plant MTFs suggested that MTFs were involved in various aspects of plant growth, development and environmental responses, such as seed germination (*Park et al., 2011*), cell division (*Kim et al., 2006*), root hair development (*Slabaugh , Held & Brandizzi, 2011*), sugar signaling (*Li et al., 2011*), ER stress (*Che et al., 2010*; *Iwata, Fedoroff & Koizumi, 2008*; *Iwata & Koizumi, 2005*; *Yang et al., 2014*), reactive oxygen species (ROS) signaling (*De Clercq et al., 2013*; *Lee et al., 2012*; *Ng et al., 2013*), cold stress (*Seo et al., 2010*), heat stress (*Gao et al., 2008*), salt stress (*Che et al., 2010*; *Kim et al., 2008*) and osmotic stress (*Kim et al., 2012*; *Yoon et al., 2008*). A further genome-wide analysis of MTFs in Arabidopsis (*Arabidopsis thaliana*) and rice (*Oryza sativa*) showed that many transcription factor families contained membrane-bound members, although most of them have yet to be functionally resolved (*Kim et al., 2010*). MTFs are fast emerging as critical drivers of gene regulation in response to various stimuli in plants. Because of their crucial role in the regulation of gene expression, it is of great importance to identify MTFs in other plant species. However, most of the MTFs that have been functionally resolved are those from Arabidopsis. This may be partially because the genome-wide analyses of MTFs have been performed using genomes of model organisms including Arabidopsis and rice. Thus, it is of great importance that a systematic analysis of MTFs is performed in various plant lineages besides Arabidopsis and rice at the genome level to not only widen the knowledge of MTFs in plant species but also provide invaluable resources for the community and further stimulate the functional elucidation of these transcription factors.

Here, we established a workflow for the identification of plant MTFs and performed genome-wide analyses of MTFs in 14 plant species spanning the chlorophytes, bryophytes, gymnosperms, monocots and eudicots. A total of 1,089 MTFs belonging to 52 gene

**Table 1  Genome information and sizes of the MTFs of analyzed plant species.**

| Species | Common name | Genome size (Mbp) | Chromosome (1N) | TF loci | MTF loci |
|---|---|---|---|---|---|
| **Chlorophyta** | | | | | |
| *Chlamydomonas reinhardtii* | Green algae | 112 | 17 | 230 | 37 |
| **Bryophyta** | | | | | |
| *Physcomitrella patens* | Moss | 480 | 27 | 1,079 | 49 |
| *Gymnosperm* | | | | | |
| *Picea abies* | Norway spruce | 19.6 Gb | 12 | 1,851 | 71 |
| **Monocot** | | | | | |
| *Brachypodium distachyon* | Brachypodium | 272 | 5 | 1,557 | 50 |
| *Oryza sativa* | Japanese rice | 372 | 12 | 1,859 | 85 |
| *Sorghum bicolor* | Sorghum | 730 | 10 | 1,826 | 62 |
| *Zea mays* | Maize | 3,000 | 10 | 2,231 | 110 |
| **Eudicot** | | | | | |
| *Glycine max* | Soybean | 979 | 20 | 3,714 | 117 |
| *Medicago truncatula* | Barrel medic | 258 | 8 | 1,577 | 71 |
| *Gossypium raimondii* | Cotton | 880 | 13 | 2,634 | 111 |
| *Solanum lycopersicum* | Tomato | 900 | 12 | 1,845 | 58 |
| *Populus trichocarpa* | Western balsam poplar | 423 | 19 | 2,455 | 89 |
| *Arabidopsis thaliana* | Thale cress | 135 | 5 | 1,716 | 64 |
| *Vitis vinifera* | Wine grape | 487 | 19 | 1,276 | 115 |

families were identified in these plant species. Most of the MTFs have one or two transmembrane motifs located in either the N- or C-terminal region, indicating that proteolytic cleavage could be a conserved mechanism for MTF activation. Additionally, a considerable proportion of the MTFs in the Arabidopsis and cotton (*Gossypium raimondii*) genome were expected to be activated by alternative splicing (ALS), indicating that ALS is another conserve regulatory mechanism for the activation of MTFs.

## MATERIALS & METHODS

### Sequence retrieval for whole membrane-bound transcription factor analysis

Complete cDNA and amino acid sequences of transcription factors of the plant species (Table 1) were retrieved from the recently published plant transcription factor database (PlantTFDB v4.0, http://planttfdb.cbi.pku.edu.cn/) (*Jin et al., 2017*). The consequent sequences were used to perform *in silico* analysis of membrane-bound proteins via the following bioinformatics tools.

### Selection of bioinformatics methods for membrane-bound protein prediction

Seven prediction methods including TMHMM 2.0 (*Krogh et al., 2001*), S-TMHMM (*Viklund & Elofsson, 2004*), HMMTOP (*Tusnady & Simon, 2001*), PHOBIUS (*Kall, Krogh & Sonnhammer, 2004*), SCAMPI-single (*Bernsel et al., 2008*), MEMSAT 1.0 (*Jones, Taylor & Thorton, 1994*) and TOPPRED (*Vonheijne, 1992*) were chosen for TM prediction. For

**Table 2** The membrane topology prediction methods used in the current study.

| Topology predictor | Algorithm | Reference |
| --- | --- | --- |
| TMHMM 2.0 | HMM | *Krogh et al. (2001)* |
| HMMTOP | HMM | *Tusnady & Simon (2001)* |
| PHOBIUS | HMM | *Kall, Krogh & Sonnhammer (2004)* |
| S-TMHMM | HMM | *Viklund & Elofsson (2004)* |
| TOPPRED | Hydrophobicity profiles | *Vonheijne (1992)* |
| SCAMPI-single | Hydrophobicity + Model | *Bernsel et al. (2008)* |
| MEMSAT 1.0 | ANN + Grammer | *Jones Taylor & Thorton (1994)* |

TMHMM 2.0, transmembrane helices and the locations were predicted; the most likely locations of transmembrane motifs were given, and they were included in this study if they had a probability score greater than 0.90. The other six predictors were employed to predict transmembrane motifs and the consequent results were filtered and/or combined by using TOPCONS-single software with the default parameters (http://single.topcons.net/) (*Hennerdal & Elofsson, 2011*).

### *In silico* analysis of alternative splicing of MTFs in Arabidopsis and cotton

Transcriptome sequences of *Arabidopsis thaliana* (TAIR10_cds_20101214_updated) were downloaded from TAIR (The Arabidopsis Information Resource) website (https://www.arabidopsis.org/). Transcriptome sequences of cotton (*G. raimondii*) were retrieved from Phytozome (https://phytozome.jgi.doe.gov/pz/portal.html#).

## RESULTS & DISCUSSION

### Establishment of workflow for identification of MTFs

To perform genome-wide analyses of the MTFs in the different plant species, seven tools that are widely used to predict transmembrane segments were selected in present study: TMHMM 2.0 (*Krogh et al., 2001*), HMMTOP (*Tusnady & Simon, 2001*), PHOBIUS (*Kall, Krogh & Sonnhammer, 2004*), S-TMHMM (*Viklund & Elofsson, 2004*), TOPPRED (*Vonheijne, 1992*), SCAMPI-single (*Bernsel et al., 2008*) and MEMSAT 1.0 (*Jones, Taylor & Thorton, 1994*). Their methods are based on different algorithms such as the hidden Markov model (HMM), the artificial neural network (ANN) and hydrophobicity (Table 2). These methods were selected because of their frequent usage in the research community and good results obtained in various studies (*Schwacke et al., 2003*; *Zhang et al., 2015*). Moreover, multiple predictors were employed to identify as many reliable membrane proteins as possible.

Sequences of Arabidopsis transcription factors were extracted from the recently published Plant Transcription Factor Database (PlantTFDB 4.0) (*Jin et al., 2017*), and they were then subjected to transmembrane region screening using the above predictors. In *A. thaliana*, for example, the genome was predicted to contain 37 to 60 membrane-bound transcription factors (MTFs) depending on the methods used (Fig. 1A). Regardless of the prediction method, most of the resulting MTFs have a single TM span, ranging from

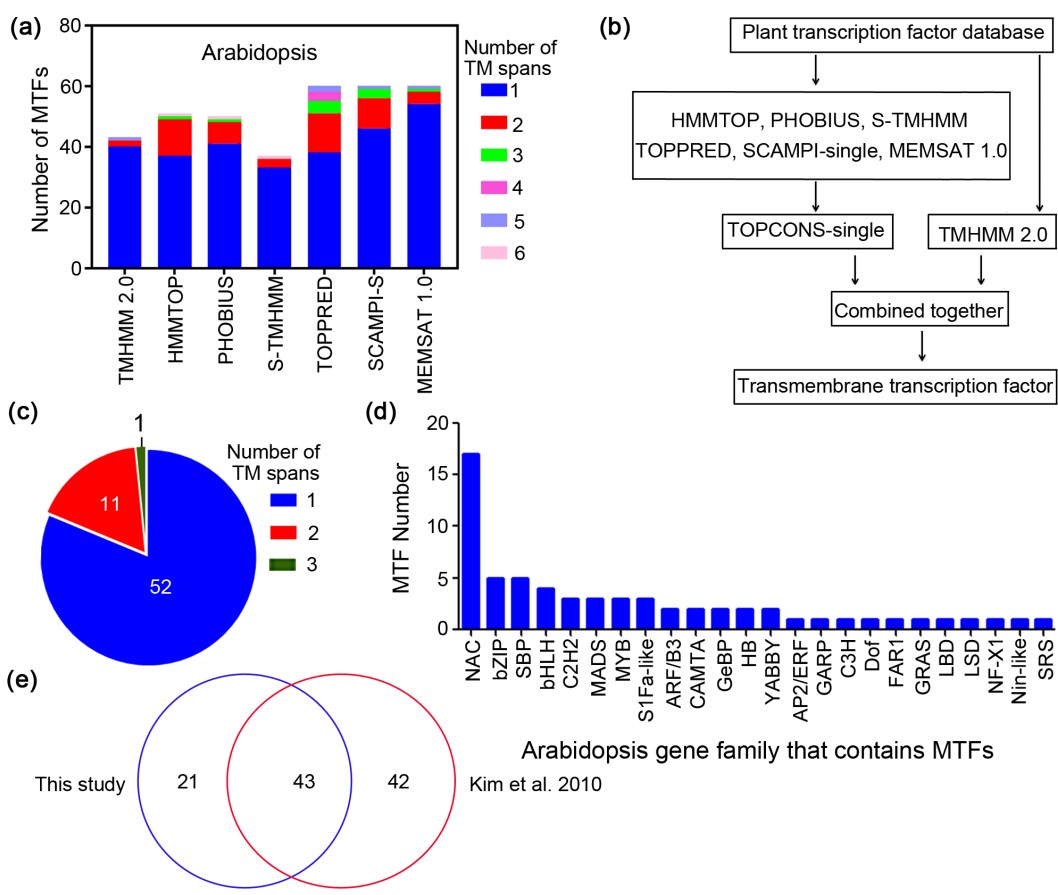

**Figure 1 Workflow for identification of membrane-bound transcription factors in plant species.** (A) Prediction of transmembrane (TM) spans of Arabidopsis by different methods; (B) workflow for identification of MTFs in plant species; (C) Arabidopsis MTFs identified in present study; (D) gene family of Arabidopsis MTFs; (E) comparison of the Arabidopsis MTFs identified in this study with the previous research.

63% to 93% (Fig. 1A). The numbers of both predicted MTFs and predicted TM motifs per protein varied among the different methods. This is because the algorithms of the methods are different; some methods may overlook membrane regions that are recognized by other methods (*Schwacke et al., 2003*). Therefore, a consensus decision was made for topology predictions based on different prediction methods, which could generate data with high accuracy and confidence (*Hennerdal & Elofsson, 2011*). The workflow for MTF identification in the present study is shown in Fig. 1B; plant transcription factor databases were subjected to analysis using the seven bioinformatics tools, including TMHMM 2.0, HMMTOP, PHOBIUS, S-TMHMM, TOPPRED, SCAMPI-single and MEMSAT 1.0. The TOPCONS-single algorithm (*Hennerdal & Elofsson, 2011*) was employed to obtain consensus prediction data for membrane proteins predicted by all of the methods except TMHMM 2.0, and the results were then combined with the data predicted by TMHMM 2.0 to constitute the membrane transcription factor database with high confidence (Fig. 1B). The results generated by TMHMM 2.0 were separately analyzed because

TMMHM 2.0 is the most accurate software currently available for membrane protein prediction, as it is characterized by a low false-positive rate but a high false-negative rate (*Schwacke et al., 2003*).

We employed this established workflow to identify membrane-bound members from the Arabidopsis transcription factors. The Arabidopsis genome was found to contain 64 MTFs (Fig. 1C), among which the number of proteins containing one, two and three membrane segments was 52, 11 and one, respectively. These proteins are classified as belonging to 24 gene families, including 17 NAC members and 5 bZIP members (Fig. 1D). The total number of MTFs identified in the present study was less than that in the previously predicted MTF database (85) in Arabidopsis (*Kim et al., 2010*), and 43 members overlapped between the two studies, which accounted for 67% of the MTFs in the present study (Fig. 1E). In contrast to the *in silico* analysis of Arabidopsis MTFs, this study identified 85 MTFs in the rice (*Oryza sativa*) genome database (Table 1), which is much more than the 45 members identified in previous research (*Kim et al., 2010*). The difference between the two studies may be due to the different algorithms used, which is common in the research on membrane protein prediction (*Schwacke et al., 2003*). However, 85 MTFs were identified in rice in the present study, while only 45 MTFs were identified in rice by *Kim et al. (2010)* (Table 1). Both the previous (*Kim et al., 2010*) and the present study have tried to identify all possible MTFs for the scientific community. One major advantage of the workflow in the present study is that these methods are freely available to the scientific community and are applicable to more plant species. The Arabidopsis MTFs identified in this study are highly reliable, because all of the Arabidopsis MTFs functionally elucidated in previous research (Table S1) were present in this study except for a PHD transcription factor (AT5G35210) that was not deposited in PlantTFDB 4.0 (*Sun et al., 2011*). These results further indicated that the workflow for MTFs identification established in this study can identify MTF with considerable reliability.

## Genome-wide *in silico* identification of MTFs in plant species

Genome-wide scale screening of MTFs has been performed in only two plant species: *Arabidopsis thaliana* and rice (*Oryza sativa*) (*Kim et al., 2010*; *Seo, 2014*). To considerably broaden this dataset and cover other plant species, we performed extensive genome-wide *in silico* analyses of various plant species as follows (Table 1). First, the sequences of plant transcription factors ranging from 230 to 3,714 loci were retrieved for different species from PlantTFDB 4.0 (*Jin et al., 2017*). Second, the transcription factor databases were screened for transmembrane motifs. Finally, a highly reliable database has been established that consists of 1,089 MTFs (Table 1). The number of MTFs in each species varied from 37 to 117 and accounted for approximately 3% to 16% of the total transcription factors. Relevant information about the MTFs, including the gene loci, transmembrane region and gene family, are shown in Table S2.

## Gene family classification of MTFs in plant species

The 1,089 MTFs identified in 14 plant species in this study belong to 52 families (Table 1). Twenty-eight additional gene families of MTFs were identified from plant species besides

**Table 3  Membrane-bound transcription factors (MTFs) in gene families of plant species.**

| Family | Plant species (MTF number) | | | | | | | | | | | | | |
| --- | --- | --- | --- | --- | --- | --- | --- | --- | --- | --- | --- | --- | --- | --- |
| Chlorophyte | | Bryo-phyte | Gymno-sperm | Monocot | | | | Eudicot | | | | | | |
| | Cr | Pp | Pa | Bd | Os | Sb | Zm | Gm | Mt | Gr | Sl | Pt | At | Vv |
| AP2 | 4 | 0 | 0 | 1 | 1 | 0 | 0 | 1 | 1 | 0 | 0 | 0 | 1 | 2 |
| ARF | 0 | 0 | 3 | 1 | 0 | 0 | 2 | 2 | 0 | 2 | 0 | 1 | 0 | 0 |
| ARR-B | 0 | 0 | 0 | 0 | 0 | 0 | 0 | 1 | 0 | 1 | 0 | 1 | 0 | 1 |
| B3 | 0 | 4 | 0 | 3 | 2 | 1 | 0 | 1 | 2 | 1 | 0 | 0 | 2 | 2 |
| BBR-BPC | 0 | 0 | 1 | 0 | 0 | 0 | 0 | 0 | 0 | 0 | 0 | 1 | 0 | 0 |
| BES1 | 0 | 1 | 1 | 0 | 0 | 0 | 1 | 1 | 0 | 1 | 0 | 1 | 0 | 0 |
| bHLH | 4 | 2 | 3 | 3 | 6 | 5 | 8 | 13 | 3 | 4 | 0 | 4 | 4 | 4 |
| bZIP | 3 | 6 | 5 | 3 | 4 | 4 | 6 | 5 | 3 | 5 | 5 | 9 | 5 | 4 |
| C2H2 | 0 | 3 | 4 | 4 | 5 | 6 | 5 | 9 | 6 | 11 | 5 | 3 | 3 | 6 |
| C3H | 3 | 2 | 7 | 3 | 3 | 0 | 2 | 2 | 3 | 2 | 0 | 5 | 1 | 5 |
| CAMTA | 0 | 0 | 0 | 0 | 0 | 0 | 2 | 2 | 2 | 5 | 0 | 0 | 2 | 0 |
| CO-like | 1 | 0 | 0 | 0 | 0 | 0 | 1 | 0 | 0 | 0 | 0 | 2 | 0 | 0 |
| CPP | 1 | 0 | 0 | 0 | 0 | 0 | 0 | 0 | 0 | 0 | 1 | 1 | 0 | 1 |
| DBB | 0 | 0 | 0 | 0 | 0 | 0 | 1 | 2 | 0 | 0 | 0 | 2 | 0 | 0 |
| Dof | 0 | 0 | 2 | 1 | 1 | 0 | 1 | 1 | 0 | 1 | 0 | 0 | 1 | 4 |
| E2F/DP | 0 | 0 | 1 | 0 | 0 | 0 | 0 | 2 | 0 | 1 | 0 | 1 | 0 | 0 |
| EIL | 0 | 0 | 0 | 0 | 1 | 0 | 0 | 0 | 1 | 0 | 0 | 0 | 0 | 0 |
| ERF | 0 | 3 | 3 | 1 | 3 | 3 | 7 | 1 | 1 | 3 | 2 | 1 | 0 | 10 |
| FAR1 | 0 | 0 | 0 | 3 | 11 | 2 | 3 | 0 | 10 | 3 | 0 | 2 | 1 | 3 |
| G2-like | 1 | 2 | 1 | 2 | 4 | 0 | 3 | 2 | 1 | 4 | 3 | 1 | 1 | 2 |
| GATA | 2 | 1 | 0 | 1 | 2 | 1 | 1 | 2 | 2 | 0 | 0 | 1 | 0 | 2 |
| GeBP | 0 | 0 | 0 | 0 | 0 | 1 | 0 | 0 | 0 | 0 | 0 | 0 | 2 | 0 |
| GRAS | 0 | 0 | 3 | 1 | 4 | 11 | 6 | 0 | 1 | 0 | 2 | 0 | 1 | 4 |
| GRF | 0 | 0 | 0 | 0 | 0 | 0 | 0 | 1 | 0 | 1 | 0 | 0 | 0 | 0 |
| HB-other | 1 | 0 | 3 | 1 | 2 | 0 | 4 | 3 | 1 | 4 | 3 | 5 | 2 | 4 |
| HD-ZIP | 0 | 1 | 0 | 1 | 0 | 1 | 6 | 3 | 0 | 6 | 0 | 1 | 0 | 3 |
| HSF | 0 | 0 | 1 | 2 | 0 | 1 | 1 | 0 | 0 | 0 | 0 | 0 | 0 | 1 |
| LBD | 0 | 2 | 1 | 0 | 0 | 0 | 0 | 0 | 1 | 1 | 1 | 0 | 1 | 1 |
| LSD | 0 | 0 | 0 | 0 | 2 | 0 | 0 | 1 | 0 | 0 | 0 | 1 | 1 | 1 |
| MIKC | 0 | 0 | 0 | 2 | 1 | 2 | 3 | 2 | 0 | 4 | 0 | 1 | 1 | 2 |
| M-type | 0 | 1 | 3 | 1 | 3 | 1 | 2 | 4 | 2 | 3 | 7 | 3 | 2 | 0 |
| MYB | 1 | 1 | 0 | 0 | 1 | 2 | 1 | 2 | 0 | 4 | 0 | 4 | 1 | 11 |
| MYB-related | 4 | 1 | 6 | 2 | 4 | 2 | 10 | 7 | 6 | 6 | 5 | 8 | 2 | 2 |
| NAC | 0 | 6 | 7 | 7 | 6 | 7 | 11 | 14 | 5 | 15 | 11 | 12 | 17 | 10 |
| NF-X1 | 1 | 1 | 0 | 0 | 1 | 1 | 1 | 2 | 1 | 1 | 1 | 1 | 1 | 2 |
| NF-YA | 0 | 0 | 1 | 0 | 1 | 0 | 1 | 1 | 1 | 0 | 0 | 0 | 0 | 0 |
| NF-YB | 0 | 0 | 1 | 0 | 0 | 0 | 0 | 3 | 0 | 1 | 2 | 0 | 0 | 1 |
| NF-YC | 0 | 1 | 1 | 1 | 1 | 0 | 0 | 0 | 1 | 0 | 1 | 0 | 0 | 0 |

| Family | Chlorophyte | Bryo-phyte | Gymno-sperm | Monocot | | | | Eudicot | | | | | | |
|---|---|---|---|---|---|---|---|---|---|---|---|---|---|---|
| | Cr | Pp | Pa | Bd | Os | Sb | Zm | Gm | Mt | Gr | Sl | Pt | At | Vv |
| Nin-like | 1 | 2 | 0 | 1 | 1 | 0 | 0 | 2 | 1 | 1 | 0 | 0 | 1 | 2 |
| RAV | 0 | 0 | 0 | 0 | 0 | 0 | 0 | 0 | 0 | 0 | 0 | 2 | 0 | 0 |
| S1Fa-like | 1 | 2 | 2 | 1 | 2 | 1 | 2 | 4 | 3 | 4 | 1 | 2 | 3 | 2 |
| SAP | 0 | 0 | 0 | 0 | 0 | 0 | 0 | 0 | 1 | 1 | 0 | 0 | 0 | 0 |
| SBP | 9 | 4 | 2 | 4 | 5 | 5 | 7 | 9 | 3 | 9 | 3 | 8 | 5 | 4 |
| SRS | 0 | 0 | 0 | 0 | 0 | 0 | 0 | 1 | 2 | 1 | 0 | 1 | 1 | 1 |
| TALE | 0 | 0 | 0 | 0 | 0 | 1 | 0 | 0 | 0 | 0 | 1 | 0 | 0 | 1 |
| TCP | 0 | 0 | 2 | 0 | 0 | 0 | 3 | 0 | 0 | 0 | 0 | 1 | 0 | 5 |
| Trihelix | 0 | 2 | 2 | 0 | 0 | 0 | 3 | 3 | 0 | 4 | 2 | 1 | 0 | 5 |
| Whirly | 0 | 0 | 0 | 0 | 0 | 0 | 0 | 0 | 0 | 0 | 0 | 0 | 0 | 1 |
| WOX | 0 | 0 | 0 | 0 | 0 | 0 | 1 | 0 | 0 | 0 | 1 | 0 | 0 | 1 |
| WRKY | 0 | 1 | 5 | 0 | 6 | 3 | 3 | 4 | 5 | 1 | 1 | 0 | 0 | 2 |
| YABBY | 0 | 0 | 0 | 0 | 0 | 1 | 2 | 3 | 0 | 0 | 0 | 2 | 2 | 0 |
| ZF-HD | 0 | 0 | 0 | 0 | 2 | 0 | 0 | 1 | 2 | 0 | 0 | 0 | 0 | 3 |

*A. thaliana* (At) (Table 3). A total of 26 families containing MTFs were identified from more than seven plant species including the AP2, B3, bHLH, bZIP, C2H2, C3H, Dof, ERF, FAR1, G2-like, GATA, GRAS, HB-other, HD-ZIP, LBD, MIKC, M-type, MYB, MYB-related, NAC, NF-X1, Nin-like, S1Fa-like, SBP, Trihelix and WRKY families (Table 3). Moreover, nine gene families of MTFs, including bHLH, C2H2, G2-like, NAC, C3H, ERF, HB-other, M-type and NF-X1, were identified from more than 12 plant species. Notably, the bZIP, MYB-related, S1Fa-like and SBP families of MTFs were identified from all 14 of the plant species. In contrast, several gene families of MTFs were identified from only a few plant species (no more than two), namely, the BBR-BPC, EIL, GeBP, GRF, RAV, SAP and Whirly families. We further summarized the total number of MTFs in the 14 species, and the top six families with the highest numbers of MTFs were as follows: NAC (128), SBP (77), C2H2 (70), MYB-related (67) and bHLH (65) (Fig. S1). These results showed that MTFs are highly diverse in various plant species. Considering that only a limited number MTFs (*Seo, 2014*), e.g., eight NAC and three bZIP transcription factors in Arabidopsis, have been functionally studied, the other MTFs deserve much more attention in further research due to the important role of MTFs in gene expression.

## Analysis of transmembrane motifs of MTFs in plant species

Among the 1,089 MTFs identified from 14 plant species, the number of TM motifs varied from one to 13. Most MTFs in the plant species, except for those in *C. reinhardtii*, have one or two TM motifs (Fig. 2A). We summarized the total number of TM regions of the MTFs in all plant species; proteins containing one and two TM motifs accounted for 71.5%, and 21.1% of the MTFs, respectively.

The location patterns of TM motifs within protein sequences were analyzed, and the major types of TM motif locations were shown in Fig. 2B. Among the MTFs containing one
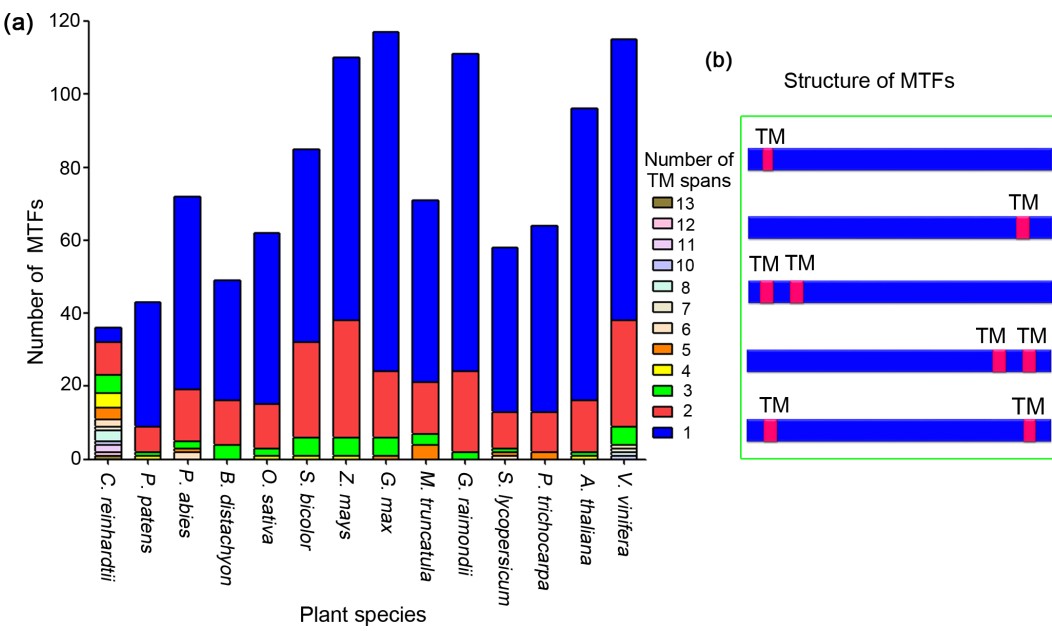

**Figure 2 Analysis of transmembrane motifs of membrane-bound transcription factors in plant species.** (A) Number of TM spans in MTFs; (B) typical locations of TM spans in MTFs. TM, transmembrane; MTF, membrane-bound transcription factor.

TM region, a total of 311 (39.9%), 360 (46.2%), and 108 (13.9%) proteins had their TM motifs in the N-terminal, C-terminal and central region of their sequences, respectively. This was slightly different from a previous study on the MTFs of *A. thaliana*, in which more than 70% members had TM motifs in their C-terminal regions (*Kim et al., 2010*). For the 230 MTF members containing two TM motifs, a total of 51 (22.2%) and 46 (20%) members had their TM regions in the N-terminal and C-terminal regions, respectively; an additional 31 MTF members (13.5%) had one TM motif located in the N-terminal region and the other motif located in the C-terminal region (Fig. 2B). Taken together, most MTFs in plant species have one or two TM domains, which are located in either their N- or C-terminal region. These results indicated that proteolytic cleavage is a conserved mechanism for MTFs in various plant species, as it is likely to release an intact functional transcription factor since the MTF was cleaved by protease near the site of the transmembrane domain. Such proteolytic process-dependent activation has been well established in Arabidopsis MTFs (*Seo, 2014*). Furthermore, the variance in TM motifs in MTFs may lead to slight differences in the details of the mechanism of MTF activation by proteolytic processing.

## Possible regulation of MTFs by alternative splicing

MTFs were once considered to be activated solely by proteolytic processing. Interestingly, recent research concerning *ZmbZIP60* and *OsbZIP74* suggested that alternative splicing (ALS) is also responsible for the activation of MTFs (*Li, Humbert & Howell, 2012*; *Lu et al., 2012*). To test whether ALS-assisted activation of MTFs was also employed by other plants' MTFs, MTF databases of Arabidopsis and cotton (*G. raimondii*) were selected, and

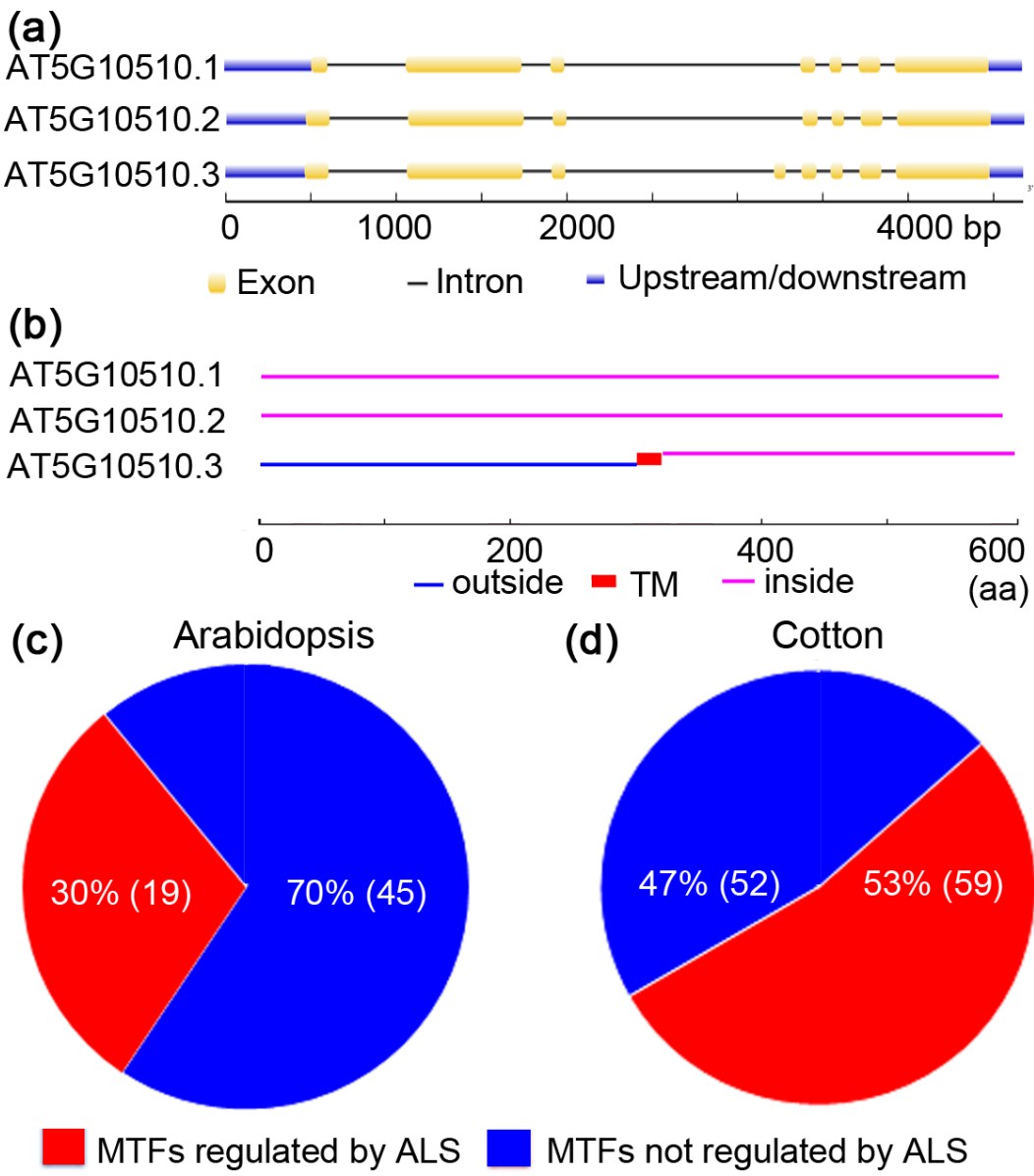

**Figure 3** **Alternative splicing of membrane-bound transcription factors in Arabidopsis (*Arabidopsis thaliana*) and cotton (*Gossypium raimondii*).** (A) Arabidopsis membrane-bound transcription factor (AT5G10510) undergoes alternative spicing (ALS); (B) alternatively spliced forms of Arabidopsis AT5G10510 lack a TM domain; (C) potential ALS dependent activation of membrane-bound transcription factors in genome of Arabidopsis and cotton.

*in silico* analyses were performed. The results are shown in Fig. 3. In an Arabidopsis MTF member (AT5G1050.3), for example, two splice variants (AT5G1050.1 and AT5G1050.2) encode proteins that maintain the nuclear transcription factor domain while deleting the TM region (Figs. 3A and 3B). Thus, ALS was proposed to be another mechanism for the activation of MTFs, in which the splice variant without the sequence encoding a TM motif would be transcribed under specific physiological conditions. In this case, the nuclear

transcription factors translocate into the nucleus and become active immediately since they were translated in the cytoplasm. The regulation of MTFs by ALS may be important in the late stage of plants in response to various stimuli. Interestingly, approximately 30% (19/64) and 53% (59/111) of genes encoding MTFs in Arabidopsis and cotton, respectively, were also shown to generate alternative splicing variants coding for proteins that only maintain the transcription factor domain with nuclear localization signal while deleting the TM motif (Figs. 3C and 3D and Table S3). Hence, these MTFs could be regulated by ALS in Arabidopsis and cotton. Moreover, ALS seems to be involved in the activation of a wide range of MTF families, such as the NAC, bHLH and MYB-related families, in Arabidopsis and cotton (Table S3). Thus, an ALS-dependent activation mechanism may also be employed by the MTFs of other plant species. However, a further "wet experiment" is needed to verify this hypothesis.

## CONCLUSION

The present study performed a systematic analysis of MTFs in plant lineages at the genome level. A total of 1,089 MTFs belonging to 52 gene families were identified in 14 plant species. Most of the MTFs have TM domains located in either the N- or C-terminal regions, indicating proteolytic cleavage could be a conserved mechanism of MTF activation in plant species. Additionally, approximately half of the MTFs in the genome of either Arabidopsis or cotton can potentially be regulated by ALS, indicating that ALS may be another conserved activation mechanism for MTFs. The present study provides invaluable information for the research community.

### Funding

This work was funded by the National Natural Science Foundation of China (Grant No. 31601520), the Technology Innovation Fund of Chongqing (Grant No. cstc2016shms-ztzx80005), the China Postdoctoral Science Foundation funded project (Grant No. 2016M592620), the Fundamental Research Funds for the Central Universities (Grant No. XDJK2016C060 and SWU115083) and the Chongqing Postdoctoral Science Foundation funded project (Grant No. Xm2016119). The funders had no role in study design, data collection and analysis, decision to publish, or preparation of the manuscript.

### Grant Disclosures

The following grant information was disclosed by the authors:
National Natural Science Foundation of China: 31601520.
Technology Innovation Fund of Chongqing: cstc2016shms-ztzx80005.
China Postdoctoral Science Foundation funded project: 2016M592620.
Fundamental Research Funds for the Central Universities: XDJK2016C060, SWU115083.
Chongqing Postdoctoral Science Foundation: Xm2016119.

## Competing Interests

The authors declare there are no competing interests.

## Author Contributions

- Shixiang Yao conceived and designed the experiments, performed the experiments, analyzed the data, contributed reagents/materials/analysis tools, wrote the paper, prepared figures and/or tables, reviewed drafts of the paper.
- Lili Deng analyzed the data, wrote the paper, prepared figures and/or tables, reviewed drafts of the paper.
- Kaifang Zeng conceived and designed the experiments, wrote the paper, reviewed drafts of the paper.

## Data Availability

The raw data is provided in Table S2.

## Supplemental Information

Supplemental information for this article can be found online at http://dx.doi.org/10.7717/peerj.4051#supplemental-information.

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
