# Peer review of "Genome-wide in silico identification of membrane-bound transcription factors in plant species"

_PeerJ, doi:10.7717/peerj.4051_

## Round 0.1 · original submission · Major Revisions

· Academic Editor

Major Revisions

Both reviewers have raised major issues in the current form of the manuscript. It is important that you address these concerns as a revision with only minor changes will not do well in the next round of review. The manuscript should be carefully edited for style and grammar.

Reviewer 1 ·

Basic reporting

1. I suggest having an English language editing service or an English-speaking colleague proofread your paper to ensure proper language usage and to improve logical flow.

2. There is sufficient background about the importance of MTFs in rapid response to stress. However, to aid the reader in interpreting the results and understanding their significance, more background is needed on the proteolytic processing of MTFs (and how this is related to the location and number of transmembrane domains), the alternative splicing of MTFs and the methods used to identify MTFs.

3. The Materials and Methods section is too brief. Most of the methods (including those presented in the Materials and Methods) are described in the Results and Discussion section, and these are not explained in sufficient detail.

4. The Titles/labels do not always accurately reflect what is shown in the figure or table, and in some cases more information is needed to interpret the figure.
-The x-axis in Fig 1d should be labeled “Arabidopsis gene families that contain MTFs”.
-In Fig 2b, I suggest providing the percentage of MTFs that have each topology.
-A more accurate title for Figure 3 would be “Alternative splicing of membrane-bound transcription factors”.
-In Fig. 3b, it is more accurate to say that alternatively spliced forms of AT5G10510 lack a TM domain. In addition, the colored lines/blocks should be explained.
-In Fig. 3c, The description “ALS-dependent activation” is not appropriate because you have not shown that these TFs are activated by alternative splicing.
-Table 1: TOPCONS-single is not included.
-Table 2 title: It is not correct to refer to these transcription factors as the “MTF family” because they belong to several different TF families.
-Table 3. The title does not reflect what is in the table, which is the number of MTFs in each TF family.

5. The TM predictions from all seven programs are not provided, and it is not clear what predictions are shown in Supplemental Table S2. Are these the combined TM predictions?

Experimental design

1. The need to identify MTFs in other species is stated (Lines 68-71), but I suggest more clearly articulating the gap in knowledge and your specific goals. For example, did you develop a new approach to identify MTFs rather than using an existing approach because there was a need for a more robust MTF identification pipeline?

2. There needs to be more discussion about the discrepancies between your results and those of Kim et al. (2010) (lines 133-134). What were the differences in your approaches that may have led to this discrepancy? You cite the fact that you identified all previously characterized MTFs as evidence that you have obtained high-confidence MTF predictions. Was this not the case for Kim et al. (2010)? If it is difficult to conclude which method is more reliable (line 134), would it be better to use both approaches when trying to identify all possible MTFs?

3. The methods need to be more fully explained.
-More detail about the TOPCONS-single algorithm (lines 117-118) and how it was implemented is needed (there is no mention of this program in the Materials and Methods). Based on Hennerdal and Elofsson (2011), only four prediction programs seem to be used by the best performing version of TOPCONS-single (http://single.topcons.net/). If this is the case, the workflow (Fig. 1) is incorrect.
-How the other prediction programs (lines 86-89) were implemented also needs to be described, including the criteria for calling a transmembrane domain.
-The reason for making TMHMM 2.0 predictions separately should be clearly stated (lines 119-120).
-Table 2 should be cited when mentioning the plant species included in the analysis (line 82).

4. The rationale for using the seven prediction programs (lines 97-101) is not fully explained. Why did you choose to use a different approach than Kim et al. (2010)?

Validity of the findings

1. The conclusion that the location of transmembrane motifs in the N- or C- terminal regions of most MTFs supports the conservation of a proteolytic mechanism for MTF activation (lines 26-28, 74-76, 192) needs to be clarified. I interpret conserved in this context to indicate evolutionary conservation, but given that the MTFs belong to different gene families, this does not seem to be correct. It is more accurate to say that the location of TMs indicates that MTFs are processed by the same proteolytic mechanism. However, why the location of the TM domains indicates this needs to be explained.

2. The conclusion that half of the MTFs in Arabidopsis and cotton could be regulated by alternative splicing (Lines 29-31, 77-79, 207) also needs clarification. It is not clear whether your conclusion that MTFs are regulated by alternative splicing is based on the presence of an alternatively spliced transcript or on the fact that these transcripts have different protein domains. In Table S3 you list the alternatively spliced transcripts and note whether they contain a TM domain, but you don’t discuss how often alternative splicing has the potential to regulate the MTF. When you say that ALS was involved in the activation of MTF families (Lines 208-209), is this based on the fact that alternative splicing removes/adds a TM domain?

3. The conclusion that MTF activation involves at least two independent mechanisms (lines 218-221) was reached by other studies. The activation of MTFs was not directly assessed in this study, and this section should be revised to specifically summarize the conclusions of your study.

Comments for the author

1. In the abstract lines (31-33) and in the conclusion (lines 215-216) you say that you have done the first systematic analysis of MTFs in plant genomes. This wording is not correct because systematic analyses have been done in Arabidopsis and rice (Kim et al 2010), which you appropriately cite in the Introduction (Lines 65-68).

2. The explanation for the discrepancies in TM prediction between algorithms is vague (lines 110-112). It would be helpful if the advantages and disadvantages of each method were explained.

3. I suggest not listing the plant names (Lines 146-150) because they are already listed in Table 2. Similarly, all of the gene families with MTFs (lines 162-169) don’t need to be listed in the text because they are already shown in Table 3.

Reviewer 2 ·

Basic reporting

no comment

Experimental design

no comment

Validity of the findings

no comment

Comments for the author

This manuscript describes the identification of membrane-bound transcription factors (MTFs) using in silico computational analysis. 1089 MTFs belonging to 52 gene family were identified from 14 plant species, most of which contain one or two transmembrane region. The authors also found that approximately half of the MTFs can be regulated by alternative splicing. The manuscript provides an important resource for readers who might be interested in MTFs in plants. However, there are several major issues that need to be addressed.

1. A previous study (Kim et al 2010) has identified MTFs from Arabidopsis and rice, and the results were quite different from the current study. 50% of the previous identified MTFs were not recovered by the current study (Fig 1 e). What could be the reason for the difference, and which method is more reliable? How many of the MTFs with known functions were identified by the current study but not by the previous study? Further analysis and discussion on this topic should be provided.

2. It is recommended to include experimental verification for some identified MTFs, for example, the 21 MTFs identified only in the current study. Arabidopsis TF resources are widely available and the experiments should be easy to carry out.

3. Do the authors have an estimation of the false discovery rate among the 1089 identified MTFs?

4. A phylogenetic analysis should be conducted to see how many of the MTFs are conserved between different plant species? Do the NAC MTFs from different species belong to the same sub-group among all NACs? How about bZIP, SBP, bHLH, and other MTFs?

5. The analysis of MTFs regulated by alternative splicing (Fig 3) depends on the accuracy of gene model annotation. While it could be assumed that the Arabidopsis gene models are quite reliable, the author should discuss how reliable are those for cotton and other plant species.

6. Table 2, Picea abies is not a Bryophyta.

7. The English language of the manuscript should be improved, such as those in lines 51 to 53.

---

## Round 0.2 · Major Revisions

· Academic Editor

Major Revisions

Reviewer 1 has raised a number of issues regarding experimental design and outcomes. In order for us to consider your manuscript, these issues should be carefully addressed. The issues raised could be addressed by text modifications and clarification. Reviewer 1 also raised issue regarding grammatical errors and sentence structure; please address these issue.

Reviewer 1 ·

Basic reporting

1. The English has been improved, but there are still grammatical errors and awkward phrasing. For example:
Pg 2 Lines 49-51. By “attached” I think you mean “located adjacent to”
Pg 2 Line 59 “been” should be deleted.
Pg 3. Line 75 “that” should be changed to “those”
Pg. 3 Line 80 “database” should be deleted.
Pg 4 Line 90 “conserve” should be “conserved”
Pg. 6 Line 153 “researches” should be “studies”
Pg. 6 Line 160 should be “workflow for MTF identification established in this study can identify MTFs”
Pg. 7 Line 176. “families of MTFs” here should be “families containing MTFs” because not all members of these families are MTFs. Similarly, in lines 186-187, it is better to say “the six families with the highest number of MTFs”
On pg. 9 line 233, “nuclear transcription factors” is not correct. Do you mean “transcription factor domain” or “nuclear localization signal” or both?

2. In the Materials and Methods the in silico analysis of alternative splicing is still not included. On pg. 4 Line 93- the table containing the list of species should be cited.
On pg. 4 Lines 106-107- what is the “the strict standard”?

3. For Supplemental Table S2 a legend that describes the contents of this table is needed.

Experimental design

1. The parameters/criteria used for membrane protein prediction are not described. If the default parameters were used, this should be stated.

2. My original confusion about how the analysis was done stems from the fact that 4 out of the 7 prediction programs in Table 1 are used by TOPCONS-single. From Hennerdal et al. 2011: “The best-performing version of TOPCONS-single, using four individual methods (Table 1), is available as an easy-to-use web-based prediction server at http://single.topcons.net/. It uses the globular protein filter of SCAMPI to weed out non-membrane proteins and then proceeds to run the rest of the predictors–HMMTOP, MEMSAT-1.0 and S-TMHMM—on the remaining set.” Thus the use of TOPCONS-single seems redundant. Why were the predictions from the 6 programs not simply combined or TOPCONS-single used directly? Note that there is a more recent reference for TOPCONS-single (Tsirigos et al. Nucleic Acids Research, 2015 6:21383).

3. On page 5 Lines 138-141, you claim that TMHMM 2.0 is the most accurate software available. However, the reference you cite (Flugge and Kunze, 2003) does not make this claim. Is the wrong reference cited?

4. I still think the rationale for using these programs needs more explanation. The points you make in your response (that you wanted a method that uses resources that are freely available to the scientific community and are applicable to more plant species) are convincing and should be mentioned in the paper. Also, you don’t explain why you used multiple algorithms, which I think was to identify as many membrane proteins as possible. Finally, you argue that these prediction programs have been frequently used and have led to good results. But the references you cite (Che et al. 2010 and Li et al. 2011) do not mention these prediction programs. References directly supporting this statement should be cited.

Validity of the findings

1. The discrepancies between your results and those of Kim et al. are still not discussed in detail. You don’t explain how the approach by Kim et al differed from yours and whether they also identified all previously known MTFs. You also don’t directly address why you found less than half of the MTFs found by Kim et al. in Arabidopsis. Do you think there are more false positive predictions in their dataset? That your datasets are complementary? What implications does this have for the discovery of MTFs in other species?
Your statement that the databases from other species are “highly reliable” (pg. 6 line 169) is not justified because you have not demonstrated that they are reliable. This does not mean that your analysis is not valid, but you should be clear about the limitations.

2. Pg 8 lines 210 and 212- The explanation for why the location of transmembrane domains indicates conservation of the proteolytic mechanism needs further clarification. For example, I think your point is that TM domains need to be located at the N- or C-terminal ends of the protein to allow a functional TF to be released by proteolytic cleavage, which occurs next to the TM domain. Are there references to back up this observation (eg. all characterized MTFs have this type of protein structure)? Does this mean that the putative MTFs with TMs located in other regions of the protein are not likely to be functional?

Reviewer 2 ·

Basic reporting

no comment

Experimental design

no comment

Validity of the findings

no comment

Comments for the author

The authors have addressed previous concerns to certain degree, though some of the points might not be resolved due to technical difficulty, such as the issue of FDR rate. As it currently stands, the manuscript is acceptable now for publication.

---

## Round 0.3 · accepted · Accept

· Academic Editor

Accept

Thank you for addressing the issues raised by Reviewer 1. The manuscript is much improved.